# Finding the Goldilocks Zone of Mechanical Loading: A Comprehensive Review of Mechanical Loading in the Prevention and Treatment of Knee Osteoarthritis

**DOI:** 10.3390/bioengineering11020110

**Published:** 2024-01-24

**Authors:** Jacob Jahn, Quinn T. Ehlen, Chun-Yuh Huang

**Affiliations:** 1University of Miami Miller School of Medicine, Miami, FL 33136, USA; tjj57@med.miami.edu (J.J.); qte1@med.miami.edu (Q.T.E.); 2Department of Biomedical Engineering, College of Engineering, University of Miami, Coral Gables, FL 33146, USA

**Keywords:** osteoarthritis, post-traumatic osteoarthritis, mechanical loading, biomechanical factors, rehabilitation exercises

## Abstract

In this review, we discuss the interaction of mechanical factors influencing knee osteoarthritis (KOA) and post-traumatic osteoarthritis (PTOA) pathogenesis. Emphasizing the importance of mechanotransduction within inflammatory responses, we discuss its capacity for being utilized and harnessed within the context of prevention and rehabilitation of osteoarthritis (OA). Additionally, we introduce a discussion on the Goldilocks zone, which describes the necessity of maintaining a balance of adequate, but not excessive mechanical loading to maintain proper knee joint health. Expanding beyond these, we synthesize findings from current literature that explore the biomechanical loading of various rehabilitation exercises, in hopes of aiding future recommendations for physicians managing KOA and PTOA and athletic training staff strategically planning athlete loads to mitigate the risk of joint injury. The integration of these concepts provides a multifactorial analysis of the contributing factors of KOA and PTOA, in order to spur further research and illuminate the potential of utilizing the body’s own physiological responses to mechanical stimuli in the management of OA.

## 1. Introduction

### 1.1. Knee Osteoarthritis

Knee osteoarthritis (KOA) represents a prevalent and formidable challenge in the management and treatment of musculoskeletal disorders and exerts a substantial impact on global health, particularly in the context of the natural aging process. The epidemiological footprint of KOA is extensive, as it emerges as one of the most common joint disorders worldwide, with models estimating a global prevalence of nearly 365 million affected individuals, with roughly 14 million of these in the United States alone [1,2]. Several studies have also noted the marked increase of KOA prevalence in elderly patients [3,4,5,6], which emphasizes the growing impact of this disease on aging individuals. The emergence of an increasingly at-risk population in terms of lifestyle, obesity, and age greatly contributes to the urgency of addressing osteoarthritis (OA) as a public health concern that demands a comprehensive foundational understanding in order to develop effective prevention and treatment strategies.

The pathophysiology of KOA highlights its degenerative nature and encompasses many intricate structural and biochemical changes within knee joints. Central to KOA is the progressive breakdown of articular cartilage in the joint over time, which normally provides cushioning and a mechanical buffer between the ends of the tibia and femur [7,8]. This process is amplified by alterations within the subchondral bone, which involves sclerosis and the formation of osteophytes [9]. Additionally, synovial inflammation further contributes to joint degradation [10]. Collectively, these pathological changes lead to the clinical manifestations of KOA, which are observed in a spectrum of symptoms that significantly impact affected individuals. Of these, knee pain, often worsened by movement, stands as a cardinal feature. Stiffness, usually after periods of inactivity, and a gradual reduction in the range of motion also contribute to the clinical presentation [11]. Importantly, these symptoms do not only affect physical well-being, but also pose a profound risk to an affected individual’s quality of life.

Various risk factors contribute to the development of KOA, which demonstrates the complexity of its etiology. Age emerges as a significant factor, with the incidence of KOA rising with advancing age [3,4,5,6]. Genetic predispositions also play a significant role in the progression of KOA, as twin studies have suggested a heritability of 50% or more [12,13]. Additionally, likely of greatest importance are mechanical factors, such as obesity, abnormal joint biomechanics or a history of traumatic injury, which impose a significant burden on joints and escalate the risk of KOA progression [14,15,16,17,18,19,20,21,22]. Furthermore, occupational stress, such as those incurred in physically demanding professions, has been demonstrated to increase the likelihood of OA development via wear and tear [23,24,25]. Thus, recognizing and mitigating these mechanical risk factors is integral to both preventative measures and the creation of personalized therapeutic approaches.

### 1.2. Post-Traumatic Osteoarthritis

Post-traumatic osteoarthritis (PTOA) emerges as a distinct clinical manifestation within the realm of osteoarthritis and is interwoven within the landscape of sports injuries and trauma of the joints. Within the realm of sports, athletes are frequently exposed to joint injury and acute mechanical loading that sets the biological stage for the development of PTOA [26,27,28]. The incidence of PTOA manifests across a wide spectrum of traumas, with a noted tenfold increase in likelihood of KOA following ligamentous or meniscal injuries [29,30]. However, there is considerable variability in the severity of PTOA observed, likely influenced by factors including the type of motion and loading stress applied, which is a primary focus of our discussion [31]. The nature of these injuries, often incurred during high-impact athletic activities, presents a unique challenge in the trajectory of PTOA development. The heightened forces and mechanical loading experienced during sports-related joint injuries contribute to a distinct pathophysiological profile of PTOA acutely, distinguishable radiologically [32], which differentiates it from primary KOA [33]. Trauma, as the primary instigator, sets in motion a series of events that alter joint mechanics and provoke long-term degenerative change [34]. Understanding the specific impact of sports-related injury and trauma on the disruption of joint health is paramount in unraveling the intricacies of PTOA development. It is still unknown how the field can use information about the inciting injury as a prognostic factor for PTOA severity.

The consequences of KOA and PTOA alike permeate beyond affected individuals alone, and greatly impact an aging, at-risk society. The impact on quality of life is profound, as influences on daily routines, recreational activities, and well-being have been observed within populations of OA patients [35]. Moreover, the socioeconomic burden associated with OA within the United States is substantial, accounting for roughly 1–2.5% of the gross national product, which encompasses healthcare costs, losses in productivity, and the strain on social support systems [36]. Addressing the multi-faceted challenges posed by OA necessitates a focused, holistic understanding of its epidemiology, pathophysiology, clinical manifestations, risk factors, and the broader implications for individuals and society alike.

In this comprehensive review, our primary objective is to delve into the intricate interplay between mechanical factors and the development, progression, and prevention of KOA, with a particular focus on PTOA. While molecular and pharmaceutical therapies targeting integral pathogenic KOA pathways have shown some promise [37,38,39], their limitations include potentially more invasive drug delivery and various side effect profiles. These limitations underscore the imperative for an increased focus on the mechanical therapy component in KOA. To do so, a deeper understanding of the impact of mechanical stress loading is necessary in order to harness these as a viable treatment option.

We compile and discuss previous studies examining mechanical loading factors in both KOA and PTOA rehabilitation. Furthermore, we navigate the experimental landscape, uncovering mechanical conditions that experimentally induce OA, in hopes of integrating these insights into our exploration of potential mechanical risk factors. The biomechanical dimensions of rehabilitation exercises take center stage, where we delve into quantitative data to unravel their impact on joint health and discuss their potential to promote OA and PTOA recovery. Through these meticulous examinations, we aim to contribute a better understanding of the biomechanical intricacies underlying OA and PTOA, paving the way for informed strategies in both treatment and prevention.

## 2. Mechanical Considerations in KOA and PTOA Pathogenesis

### 2.1. Mechanotransduction

To begin, we delve into the biological underpinnings of mechanotransduction, exploring the pathways through which mechanical forces translate into biological responses. The knee joint, in its healthy state, functions as a hinge joint, which enables vital movements like flexion and extension through the interaction of articular cartilage, synovial fluid, ligaments, and menisci that allow for smooth and continuous motion of the joint. Articular cartilage, the resilient tissue covering the ends of bones, distributes mechanical forces in order to promote frictionless movement. Ligaments provide stability, while the synovial fluid lubricates and nourishes the joint. Together, these components synergize to coordinate the typical physiological processes within the knee joint that allow for seamless motion and weight-bearing activities.

In the intricate milieu of KOA, the role of mechanical loading arises as a pivotal determinant in both the pathogenesis and recovery of the knee joint and is intricately linked to mechanotransduction processes. Mechanotransduction, described as the conversion of mechanical signals into biochemical responses, is paramount to understanding how mechanical loading influences the fate of articular cartilage. In the context of this review, it is critical to note that KOA pathogenesis is deeply involved with aberrations in mechanical loading, which disrupt the delicate balance of loading force required to maintain joint health. In the widely accepted pathological scenario [40,41,42,43,44,45], supramaximal mechanical forces act as a “trigger” for maladaptive responses in chondrocytes, the cells within articular cartilage [46]. Research conducted by Buckwalter and colleagues sheds light on the intricate relationship between mechanical loading and articular cartilage fate [46]. Their research emphasizes that both acute impact events and cumulative contact stress initiate the release of reactive oxygen species from mitochondria, leading to chondrocyte death and matrix degradation. Importantly, the study illuminates a substantial difference between PTOA primarily caused by acute intense joint injury and OA resulting from chronic joint instability or incongruity. Furthermore, the study describes the capacity of joints with advanced PTOA to remodel and improve with appropriate treatment, emphasizing the dynamic nature of the mechanotransduction processes involved in joint health and repair.

There is ample evidence in support of Buckwalter’s research [44,47,48,49,50,51], as it is well characterized that high-impact forces exerted on the joint lead to maladaptive responses in chondrocytes and subchondral bone. As a result, chondrocytes may exhibit increased production of matrix-degrading enzymes, leading to cartilage breakdown and the activation of several inflammatory pathway cascades [38,52,53]. In conjunction with their pathogenic effects, mechanotransduction pathways activated by beneficial or appropriate exercise and mechanical forces stimulate the synthesis of essential extracellular matrix components, which fosters an environment that is favorable to tissue repair and injury prevention [44,54,55,56,57,58,59]. While promising, the linkage between mechanical loading and KOA progression requires further investigation as a therapeutic option to understand how mechanical loading can induce beneficial regenerative responses in chondrocytes [60]. Through this interplay, well-regulated mechanical loading emerges as a key player in the preservation and restoration of joint health in the context of KOA and PTOA.

### 2.2. Mechanical Factors in PTOA

The mechanical aspects of PTOA pathogenesis involve complex interactions between joint biomechanics, tissue biology, and the body’s capacity for repair and maintenance of joint health. For example, disrupted joint biomechanics, as a result of trauma or sports injury, will disrupt the delicate balance surrounding mechanotransduction pathways. Altering either cartilage synthesis or degradation creates a mismatch that contributes to accelerated joint degeneration [61,62]. Additionally, progressive changes in load distribution within the joint due to aging or injury, whether it be from misalignment or other contributing factors, can result in abnormal stresses, leading to the breakdown of cartilage and inflammatory responses, as seen in PTOA and KOA alike [54,63,64]. Research has also elucidated several variables that contribute to PTOA progression, such as variations in flexion angles and resistance [29,30,65]. Therefore, it is crucial to understand that PTOA is complex and under active investigation.

Mechanical factors, such as the type of motion and loading stress incurred by the joint, significantly influence the severity of PTOA. Previous studies shed light on the complicated relationship between loading stress and PTOA progression. A previous study, utilizing rabbit models subjected to different impact loads, demonstrated that articular cartilage could tolerate single impact loads up to 45% of the joint fracture threshold without significant disruption or degradation [66]. These findings emphasize not only the resilience of articular cartilage, but also highlight the potential long-term consequences of acute mechanical injury and provide a reference point for quantifying “excessive” or supramaximal joint loading.

Previous studies by D’Lima et al. [67,68], focusing on knee joint forces and their impact on OA, underscore the critical role of factors like body weight, muscle contractions, and biomechanics in influencing knee forces. The research emphasizes that each additional kilogram in body weight is multiplied two or three times at the knee, contributing to increased joint loading and potentially accelerating arthritis progression. Malalignment of the lower extremity, as previously discussed, is also identified as a factor associated with the progression of osteoarthritis, which is thoroughly characterized [69,70]. D’Lima et al. [67] also examined knee forces during exercise and recreational activities after knee arthroplasty, providing valuable insights applicable to PTOA. Their study demonstrated, like others [71,72,73], that activities such as running, golf, and tennis were found to produce unexpectedly high forces, especially in the leading knee, emphasizing the need for careful consideration of post-traumatic joint health in athletes engaging in such activities. While the correlation between the increased forces incurred during high-load activities and KOA is unclear [74,75,76], it is evident that these activities do predispose an individual to injury of the knee [77,78], which can contribute to OA progression.

Furthermore, investigation into contact stresses in the knee joint during deep flexion activities by Thambyah and colleagues revealed significantly higher peak stresses, especially in the medial compartment, during squatting, a common resistance exercise [79]. The study raised concerns about the adequacy of articular cartilage to support high contact stresses during deep flexion and exemplifies the need to consider mechanical factors in common exercises as contributors to the development of PTOA.

Wallace and colleagues quantified patellofemoral joint reaction forces and stress during the squat maneuver and found that patellofemoral joint stress increases linearly with increasing knee flexion angle and joint force [80]. The addition of external resistance further elevated patellofemoral joint reaction force and stress. The study suggested that limiting terminal joint flexion angles and resistance loads could help minimize patellofemoral joint stress during squatting activities and may hinder osteoarthritis progression. These insights, among those previously discussed, emphasize the importance of considering specific motions and loading conditions in understanding the mechanical factors influencing the severity of PTOA, and exemplify the multifactorial nature of OA progression (Table 1).

Prior research, employing cadaver, in vivo, and in vitro animal cartilage, revealed that chondrocyte death can be triggered by impact stress as low as 18 MPa (megapascals). Furthermore, impact stress exceeding 30 MPa was observed to inflict surface damage on cartilage, ultimately contributing to cartilage degradation [34,81,82,83,84,85,86,87]. Patellofemoral joint stress linearly correlates with joint force [80], with the ratio of patellofemoral joint stress to force being about 2.3 MPa per body weight according to previous studies [79,88]. It suggests that exceeding eight times the body weight in patellofemoral joint forces may lead to potential cellular injury in articular cartilage by surpassing the critical threshold of 18 MPa. Joint forces are commonly quantified in terms of body weight (BW) in exercises (see next section) and can be used as a quantitative measure to evaluate various joint loading conditions to mitigate or prevent OA.

**Table 1 bioengineering-11-00110-t001:** Literature Review: Current research on mechanical factors contributing to PTOA and KOA progression, the therapeutic potential of mechanical loading, and quantification of mechanical forces in rehabilitation exercises that were utilized within this review.

Study Focus	Author, Year [Source]	Study Type	Number of Patients	Study Description
Impact of Athletics on Development of PTOA	Hootman et al., 2007 [26]	Epidemiological	182,000	Summarizing injury data to identify preventable risk factors for injury prevention strategies
	Golightly et al., 2009 [27]	Epidemiological	2528	Describes prevalence of KOA within retired football players
	Drawer et al., 2001 [28]	Epidemiological	500	Determines the prevalence of KOA within retired soccer players
	Kujala et al., 1995 [31]	Epidemiological	117	Analyzing the impact of increased mechanical loading during sport on KOA
	Swärd et al., 2010 [32]	Epidemiological	331	Compares radiographic structural changes of KOA and PTOA in athletes v non-athletes
	Boocock et al., 2009 [72]	Epidemiological	20	Investigates the effect of running on cartilage degeneration in athletes
	Thelin et al., 2005 [78]	Epidemiological	825	Analyzes the risk of KOA development in patients with sports participation and/or injury
	Shaw et al., 2004 [89]	Epidemiological	258	Evaluates the effect of triathlon training on likelihood of future injury
	Koplan et al., 1995 [90]	Epidemiological	535	Analyzes the impact of exercise on risk of knee injury and KOA development
	Piggot et al., 2009 [91]	Epidemiological	16	Analyzes the relationship between training load and injury in football players
	Satterthwaite 1999 [92]	Epidemiological	875	Investigates the impact of marathon running on prevalence of injuries in athletes
	Clausen et al., 2015 [93]	Epidemiological	326	Investigates the effect of previous knee injury on risk of future knee injury in soccer players
Mechanical Contributors to OA	Gillquist et al., 1999 [29]	Literature review	-	Summarizes the risk of ligamentous injury for osteoarthritis progression
	Hunter et al., 2005 [63]	Epidemiological	162	Analyzes the effect of malalignment in KOA progression
	Timmins et al., 2017 [75]	Systematic review	-	Determines the effect of running on development of KOA
	Schueller-Weidekamm et al., 2006 [94]	Epidemiological	26	Analyzes the long-term changes of the knee via MRI in former long-distance runners
	Bosomworth 2009 [76]	Systematic review	-	Analyzes the effect of exercise on risk of KOA
	D’Lima et al., 2008 [67]	Epidemiological	3	Investigates the effects of various activities on mechanical loading in the knee joint
	D’Lima 2006 [95]	Literature review	-	Synthesizes studies characterizing forces on the knee joint during various exercises
	Borelli et al., 2004 [66]	In vivo	-	Investigates the impact of varied loading conditions on knee cartilage in rabbit models
	Whittaker et al., 2022 [65]	Systematic review	-	Analyzes the effect of previous knee injury on PTOA progression
	Felson et al., 2013 [64]	Epidemiological	11,006	Characterizes the effect of malalignment on KOA progression
	Aljehani et al., 2022 [96]	Epidemiological	229	Investigates biomechanical predictors of KOA progression
	Driban et al., 2015 [97]	Epidemiological	4435	Analyzes the impact of knee pain or previous injury on the likelihood of future injury
	Lieberthal et al., 2015 [61]	Systematic review	-	Compiles evidence for the role of inflammation in joint injury and PTOA
	September et al., 2007 [98]	Literature review	-	Discusses risk factors and contributing elements to dysfunction in several joints
	He et al., 2020 [51]	In vivo	-	Evaluate effect of lessened mechanical loading on KOA in mouse model
	Fang et al., 2020 [52]	Systematic review	-	Summarize the biological underpinnings of the effect mechanical loading on KOA
	Schroder et al., 2019 [50]	In vitro *	5	Investigates the impact of mechanical loading on gene expression within chondrocytes in OA and non-OA patient samples
	Sharma et al., 2001 [69]	Epidemiological	237	Investigates the impact of alignment on KOA progression
	Tanamas et al., 2009 [70]	Systematic review	-	Analyzes the correlation between malalignment and KOA progression
	Neelapala et al., 2020 [99]	Systematic review	-	Summarizes current evidence on the effect of hip muscle weakness in KOA patients
	Zhu et al., 2020 [47]	In vivo	-	Analyzes the effect of mechanical loading on subchondral bone, cartilage, and KOA
	Robbins et al., 2011 [42]	Epidemiological	38	Evaluates the effect of increased mechanical loading of functional scores of KOA patients
	Milentijevic 2005 [34]	In vivo	-	Investigates the impact of loading stress on rabbit articular cartilage
	Roos et al., 1998 [30]	Epidemiological	123	Determining the effect of meniscal surgery/removal on osteoarthritis progression
Therapeutic Potential of Mechanical Loading	Frontera et al., 1988 [56]	Epidemiological	12	Analyzing the impact of strength-training regimen on muscle development
	Nebelung et al., 2012 [58]	In vitro *	8	Investigates gene expression of human chondrocytes following total knee replacement
	Veugelers et al., 2016 [100]	Epidemiological	45	Analyzes the impact of varied training loads on risk of future injury
	Baert et al., 2014 [101]	Systematic review	-	Investigates the impact of lateral wedge insoles in patients with KOA
	Fantini-Pagani 2011 [102]	Epidemiological	10	Analyzes the benefit of knee brace on minimizing force at the knee joint
	Robert-Lachaine 2022 [103]	Epidemiological	10	Investigates the impact of knee brace and orthoses in KOA treatment
	Barrios et al., 2010 [104]	Epidemiological	8	Investigates the effect of malalignment on KOA progression
	Jan et al., 2008 [105]	Epidemiological	102	Compares the effects of high and low load training regimens on KOA functional scores
	Kunduracilar 2018 [106]	Epidemiological	89	Investigates the effect of water training (low load bearing) on KOA progression
	Tagesson et al., 2008 [107]	Epidemiological	42	Analyzes the effect of quadriceps strengthening on KOA functional scores
	Heywood et al., 2019 [108]	Epidemiological	41	Analyzes variance in water v. land conditions while performing various exercises in KOA
	Vleck et al., 2010 [109]	Epidemiological	35	Investigates the effect of varied training regimens on risk of future injury
	Soligard et al., 2016 [110]	Literature review	-	Summarizes the impact of loading conditions on risk of injury, and injury prevention
Quantification of Mechanical Forces in Rehabilitation Exercises	Thambyah et al., 2005 [79]	In vitro *	5	Quantifies mechanical forces present at knee joint during walking
	Wong et al., 2011 [111]	In vitro *	4	Analyzes mechanical forces present on knee joint during various malalignment conditions
	Sasaki 2010 [112]	Computer model	-	Utilizes walking simulation to determine the muscles involved in walking
	Holyoak et al., 2019 [113]	In vivo	-	Quantifies mechanical forces during compression to elucidate beneficial loading range
	Glass et al., 2010 [114]	Systematic review	-	Analyzes the mechanical loading force of OKC and CKC exercises at the knee joint
	Bini 2017 [115]	Computer models	-	Utilizes computer simulated models to analyze forces on the knee joint during leg extension
	Escamilla et al., 1998 [116]	Epidemiological	10	Quantifies mechanical force at the knee joint during squat, knee extension, and leg press
	Escamilla 2001 [117]	Literature review	-	Discusses the mechanical forces incurred at the knee joint during the squat
	Schoenfeld 2020 [118]	Literature review	-	Summarizes the mechanical load incurred during the squat
	Perez et al., 2015 [119]	Literature review	-	Summarizes the mechanical load during various rehabilitation exercises
	Wallace et al., 2002 [80]	Epidemiological	15	Quantifies patellofemoral joint forces during the squat

***** Denotes that in vitro samples were conducted on clinical patient samples.

### 2.3. The Goldilocks Zone

Establishing a Goldilocks zone of loading that preserves joint health while preventing or mitigating joint injury is an essential step for developing strategies to prevent and treat PTOA (Figure 1). As delineated by Sokoloff’s excellent aphorism in 1969 [120], “cartilage can survive in a large range of solicitations, but below or beyond, it will suffer”, illuminating the ideal range of mechanical loading is pivotal for optimizing joint health and the management of PTOA. Studies have described, in-depth and through varied language [121,122], what we are referring to as the Goldilocks zone. A 2016 International Olympic Committee (IOC) publication [110] critically analyzed the effects of underloading and overloading on athlete injury and performance. Within their discussion, the IOC cites evidence demonstrating the counterbalance between the increased risk of injury in athletes training with sub-competition loads [89,91,92,109], and the increased risk of injury with sustained, high-intensity loads [26,90,94,123,124]. Interestingly, however, the IOC also cites evidence demonstrating a beneficial effect of high-intensity loading on injury prevention [91,100,109,125]. This, in essence, demonstrates that high-intensity loading in the Goldilocks zone could prevent joint injury and damage, whereas exhibiting opposite effects outside the zone. Therefore, integrating the current knowledge from mechanobiology studies of knee joints into rehabilitation programs allows us to define the Goldilocks zone as a guide for the development of targeted exercises that optimize beneficial mechanical forces, fostering tissue repair and functional improvement.

This schematic depicts the factors of mechanical load that can stimulate the knee joint. On one end of the spectrum, less frequent repetitive movements performed over long periods of time, such as standing or slow walking with inadequate biomechanics, abnormal joint loading (injury) or excess load (obesity), can cause chronic degradation resulting in OA [44]. On the other end of the spectrum, acute movements with high force can cause acute injury and induce PTOA. Movements that occur with minimal load, regardless of frequency, do not provide an adequate stimulus to induce chondral regeneration. We propose that in the middle lies a Goldilocks zone (in green color) with therapeutic potential. Applying a mechanical load ranging from approximately 0.25 BW to 8 BW at a frequency range of 0.1–3 Hz and with proper joint biomechanics can be used to induce chondral regeneration without risk of further injury [119,126,127].

## 3. Biomechanics of Rehabilitation Exercises

### 3.1. Rehabilitation Techniques

For patients struggling with OA or recovering from an injury in hopes of preventing PTOA, physicians often use physical therapy as a first-line treatment option. Physical therapy utilizes a variety of techniques and exercises to reduce pain, stimulate blood flow and healing, recover strength and range of motion, and promote functionality. Some techniques include electrical or ultrasound stimulation, heat or cold therapy, massage, or alternative modalities such as acupuncture. The exercises used are often progressive in nature, continuously working to increase the range of motion or add resistance. The biomechanics of these rehabilitation exercises have been extensively studied, describing many physical properties. Metrics that are often analyzed include muscle activation, maximal force exerted on a joint, shearing force, impulse, and other variables, the most common of which can be seen in Figure 2. It is the understanding of these physical measurements that connect the clinical data to the translational pre-clinical models in which the same metrics have been proven to promote biochemical changes. Therefore, by studying the biomechanics of rehabilitation, physical therapy exercises and techniques can be optimized to administer a therapeutic mechanical load that creates the most suitable environment to stimulate joint healing.

Improper biomechanics, influenced by the mechanical factors discussed previously, are known to be the culprit in certain musculoskeletal pathology. Therefore, through an understanding of improper biomechanics, many studies have found that the use of proper mechanics in rehabilitation exercises is an effective modality of promoting joint healing and functional recovery. These rehabilitation exercises can be broken down into two main categories: resistance exercises and aerobic exercises, both of which are explored in greater detail below.

This schematic depicts the many components that can modify the mechanical stimulus of the knee, which impacts the body’s physiologic response. (1) Mechanical load is the force of gravity +/− external load. (2) Knee flexion angle is the angle between the femur and tibia. (3) Knee adduction angle is the external rotation of the tibia relative to the femur. (4) Shearing force is a result of anterior/posterior translation of the tibia relative to the femur. (5) Tibiofemoral compression is the force felt inside the joint as a result of external forces, and joint-spanning muscle contractions. (6) Patellofemoral compression is the force of the patella pushing against the femoral condyles.

### 3.2. Resistance Exercises

The main function of resistance exercises is to build muscle strength. Therefore, resistance-based rehabilitation is focused on regaining strength in the joint and building up type II muscle that can produce maximal force [128]. Using resistance is also a modality to deliver a desired compressive force to bones and joints in hopes of generating a remodeling stimulus. Adequate muscle strength is imperative for prevention and symptomatic relief of OA. For example, quadriceps weakness has been shown to elicit more pain in patients with OA [129]. Strengthening the quadriceps has proven to be joint protective and decrease the compression of the tibiofemoral compartment through adequate ground reaction forces. Hip abductor weakness is proven to cause a shift in the center of mass, creating knee adduction motion while walking and contributing to medial compartment OA [99]. Strengthening the hip abductors has also been proven to reduce pain. Finally, compressive loading at the knee joint is not only limited to the muscles spanning the knee. Studies have shown that other muscles such as the gluteus medius and soleus contribute to tibiofemoral compression through force transfer and ensuring proper biomechanics [129]. It is for these reasons that resistance exercises and muscle strengthening are essential for the prevention and treatment of OA.

The resistance exercises used to strengthen muscle can be further categorized based on resistance type and mechanism of movement. The resistance type includes either BW or loaded exercise. The load can be induced with free weights (e.g., barbell/dumbbell) or other resistance methods such as banded exercises, which increase resistance as the movement progresses. The loaded effect can be modified in many ways to achieve the desired maximal force on the joint and surrounding muscles.

Exercises can also be categorized by the mechanism of movement, including closed kinetic chain (CKC), open kinetic chain (OKC), and isometric exercises. CKC indicates that the distal point of the extremity is fixed in place while the joint flexes and extends with motion in multiple axes, as in a squat or leg press. An OKC exercise utilizes a free-moving distal extremity when the joint flexes or extends. It occurs in one primary axis and places rotary stress on the joint [130,131]. Examples of this include leg extensions or leg curls, in which the knee itself does not translate in space, but the tibia and foot either extend or flex to open and close the knee joint. Where CKC exercises cause co-contraction of the agonist and antagonist muscle groups, the OKC mechanism uses an isolated movement to target a specific muscle group [130,131]. Both of these movement mechanisms consist of an eccentric and concentric component which refers to muscle contraction while lengthening and shortening, respectively [132]. For example, when one performs a leg extension, their quadriceps contract and shorten the muscles as a concentric movement. Lastly, isometric movements include muscles contracting against a fixed object without the joint angle changing. Both extending and flexing movements can use an isometric technique. Isometric techniques are often used to measure maximal force able to be produced by a muscle group.

### 3.3. Squat

The squat is a CKC exercise that has great translatability to both athletic performance and everyday tasks such as standing up from a seated position and climbing stairs. The movement recruits the quadriceps femoris, including the vastii and rectus femoris, as well as the gluteus maximus, hip adductors and abductors, and hamstrings. It is estimated that performing a squat activates over 200 muscles in the body [118]. During the squat descent, the quadriceps eccentrically contract, with peak activation at 80–90 degrees, while the hamstrings co-contract to stabilize the knee [118]. The hamstring contraction reduces stress on the ACL and decreases shear force, which is often why it is considered a safer option for rehabilitation than OKC options. However, this co-contraction also increases tibiofemoral compression force. The compressive force as well as posterior shear force is highest at peak flexion at the bottom of the exercise movement. Conversely, the maximum anterior shear force is observed during the first 60 degrees of flexion. Patellofemoral compression describes the force between the patella and femoral condyles. Similar to the tibiofemoral compression force, patellofemoral compression is maximal at the bottom of the movement [118]. During the ascent, the quadriceps and gluteus maximus contract concentrically and the exercise is finished in the standing position.

Many factors can be modified to target different muscles and change the forces felt by the knee. First, external load can be added, often in the form of dumbbells or a barbell. Adding load does not change the shear force and therefore does not change the risk of ACL or posterior cruciate ligament (PCL) injury [116]. However, it does add significantly more compressive load to the tibiofemoral joint. Another factor is the width of the stance when squatting. A wider stance decreases the compressive force at the tibiofemoral joint and also decreases shear force while recruiting more gluteus maximus and adductor longus activation. Conversely, a narrow stance increases the compressive force at the knee joint and recruits more gastrocnemius activation [118]. The depth of the squat can be altered to target certain muscles as well. For example, one looking for quadriceps strengthening with minimal compressive load may want to stop their descent between 80–90 degrees of flexion [117]. Quadriceps activation does not increase with further depth, while compressive forces continue to increase. Finally, the speed of the squat impacts the forces felt by the knee. A faster squat causes higher shear and compressive forces compared to a slow and controlled squat. While athletes may be training for explosive movements, someone looking to prevent OA should perform slow and controlled squatting to minimize excess compressive and shearing forces [118].

The descending and ascending portions of squatting emulate descending and ascending stairs, respectively. Kutzner et al. measured the compressive forces during these movements and found that descending and ascending stairs produced compressive forces of 3.46 BW and 3.16 BW, respectively. Standing up and sitting down with both legs produced compressive forces of 2.46 BW and 2.25 BW, respectively [133]. Overall, the total force that can be applied to the knee joint during squatting has been proven to range from 2.44 BW to 7.3 BW depending on load and knee flexion angle (Table 2) [79,80,88,119,134].

### 3.4. Leg Extension

The leg extension is an OKC exercise that is widely used for quadriceps strengthening. The exercise begins with the knee in a flexed position, and, as the quadriceps concentrically contract, the leg is extended to almost 0 degrees flexion. The quadriceps then eccentrically contract, allowing the resistance to pull the leg back into flexion. It is important to note that during the knee extension, the tibial plateau is typically translated anteriorly when approaching complete extension. While this is unproblematic in a structurally sound knee, it must be considered when rehabbing an ACL injury as it may put unwanted strain on the ACL ligament [135].

During the course of the exercise, the compressive forces placed on the tibiofemoral joint are equal to the patellar tendon force. Essentially, because there is no body weight component in this seated exercise, the only compression is caused by the quadriceps muscle contraction itself. This compressive force peaks at 60 degrees of flexion and was shown to range from 7000 N to 8500 N depending on the load and tibial length [115]. Alternatively, the shear force present at the knee joint is posterior in the full flexion starting position and transitions to maximum anterior shear force at full extension, estimated to be between 200 N and 900 N depending on the load and tibial length [115,135]. Overall, the total force that can be applied to the knee joint during the leg extension has been proven to range from 1 BW to 5.04 BW depending on the load (Table 2) [67,119,136].

### 3.5. Comparison

OKC and CKC place different magnitudes of tension on different muscles throughout the activity, resulting in variable forces on the joint. Escamilla et al. performed an extensive study analyzing the different forces on the knee joint during the squat, leg press, and knee extension [116]. They found that the muscles recruited were not only a function of the stage of motion in the exercise (flexion vs. extension), but also the type of exercise that was being conducted. For example, during the leg extension, the rectus femoris was most activated while undergoing flexion from 15° to 65° and extension from 57° to 15°. However, the rectus femoris had even more activation from 83° to 95° with the CKC exercises. This finding elicits the possibility of targeting specific muscles from two modalities: exercise-specific and degree of flexion/extension. Similarly, the forces placed on the joint were also a function of exercise and degree of flexion/extension. The OKC exercise placed minimal compression force on the joint, despite having higher shear force and extension torque at low degrees of flexion compared to the CKC exercises [116]. The takeaway from this finding is twofold. OKC can be used to target specific muscle growth (e.g., quadriceps), which can improve joint stability and potentially prevent the progression of OA. However, OKC exercises do not provide mechanical compression to the joint, which is often the suspected stimulus required for chondral regeneration, thus limiting their usefulness for OA therapy.

There is some conflicting evidence present in the literature about the efficacy and safety of OKC exercises. Contradictory to the previous findings, Glass et al. support the use of both OKC and CKC, stating that there is no significant difference in joint laxity or anterior tibial translation [114]. This was supported in a biomechanical analysis of ACL-deficient patients, which found that OKC exercises had no difference in tibial translation during movement but resulted in significantly higher quadriceps strength compared to CKC exercises [107]. Further studies must be conducted that not only measure anterior translation and quadriceps strength but include other important metrics such as shear force and compressive force that are consequential in the healing of knee joints.

### 3.6. Aerobic Exercises

Contrary to resistance-based rehabilitation, aerobic exercises are less about building strength and more focused on restoring function by moving the body repetitively in natural states of motion. Initially, modalities often include walking, jogging, and swimming. From this baseline, athletes can build up to plyometrics, like jumping and cutting. Aerobic rehabilitation includes reconditioning muscles with type I muscle fibers, preparing the muscles for long-term use for joint stabilization [128]. Aerobic exercise also provides repetitive compression at the joint. In a mouse model, Holyoak et al. describes the application of cyclical tibial compression as a way to attenuate cartilage degradation and osteophyte formation following joint trauma [113]. Importantly, the study exhibited that early initiation of mechanical compression following injury yielded histological evidence of OA prevention, providing a basis for utilizing mechanical loading to advantageously prevent microscopic pathological changes. Further studies must be conducted to optimize the time to start aerobic rehabilitation and the magnitude of compression to best attenuate the progression of PTOA.

Walking provides a different stimulus to the joint and surrounding muscles compared to resistance-based exercises. Ambulation can be broken down into two stances, an early and late stance. The early stance is when one’s foot plants on the ground and, at this point, the vastii are the main muscle contributors and contract to stop the foot and slow down the body. Here, the average tibiofemoral joint force is approximately 2.8 BW [112]. As one shifts their weight forward into the late stance, the gastrocnemius becomes the primary contributor, pushing off the ground and generating an average tibiofemoral joint force of approximately 1.9 BW [112]. These compressive forces are similar to those of a one-legged stance, which produces a force of 2.59 BW [133]. In addition to body weight, the amount of force felt by one’s knee is also dependent on their biomechanics. In the aforementioned study by Rooney and colleagues, analyzing the difference between forefoot striking and rearfoot striking while jogging, it was determined that forefoot striking had a significantly higher force on the ankle (41.7%) and knee (14.4%) relative to rearfoot striking [137]. Another contributing factor is the activation of other leg muscles such as the gluteus maximus and soleus, which react and absorb ground force [112]. Overall, the total force that can be applied to the knee joint during aerobic exercises depends on the exercise and load. Knee forces when walking, cycling, using the elliptical, and climbing stairs has been proven to range from 1.0 BW to 4.0 BW [79,95]. Knee forces when jogging or running typically ranges from 3 BW to 4.2 BW [67]. Finally, knee forces when jumping and landing ranges from 6.7 BW to 10.4 BW (Table 2) [119,138,139]. Based on this knowledge, proper patient education and biomechanical analysis can be used to create optimal knee compression in patients with OA. Both resistance-based and aerobic rehabilitation are necessary to stabilize the joint, increase functionality, and improve the patient’s quality of life.

**Table 2 bioengineering-11-00110-t002:** Mechanical forces during rehabilitation exercises: This table summarizes the tibiofemoral forces experienced during closed chain and open chain kinetic exercises (CKC; OKC). Additionally, a range of loading forces is provided for various aerobic exercises.

Type	Exercise	Tibiofemoral Force	Reference
CKC	Squat	2.44–7.3 BW	[79,80,88,134]
	Leg press	3.0–6.32 BW	[67,136]
OKC	Leg extension	1.0–5.04 BW	[67,136]
Aerobic	Walking	1.0–3.5 BW	[95,140]
	Elliptical	3.0–4.0 BW	[67]
	Stairmaster	3.0–4.0 BW	[67]
	Jogging/running	3.0–5.1 BW	[67,140]
	Jumping	6.7–10.4 BW	[138,139]

Adapted from Perez et al., 2015 [119].

### 3.7. Mechanical Loading as a Therapeutic for KOA

The purpose of the biomechanical analyses in the previous section is to understand the positive and negative attributes of common rehabilitation exercises, how they distribute force onto the joint, and their potential impact on OA progression or treatment. There have been many in vitro and in vivo mechanobiology studies that investigated potential therapeutic mechanical loads to treat OA.

Dynamic compression at physiological levels with the frequencies of 1–2 Hz in vitro has been found to induce chondrocyte proliferation and upregulate glycosaminoglycan synthesis as well as other components of the extracellular matrix, whereas static compression suppresses the metabolic activity of chondrocytes [44,59]. Furthermore, several in vitro investigations, including research conducted in our laboratory, have illustrated that subjecting chondrocytes to dynamic compression at moderate levels can impede their catabolic and apoptotic responses to mechanical injury and/or pro-inflammatory cytokine challenges [141,142,143,144]. Our recent ex vivo study devised a novel dynamic joint loading system to closely and clinically mimic in vitro dynamic compression. This system combines compressive tibial axial loading with continuous passive motion and was shown to decrease the expression of inflammatory cytokines within articular cartilage in an ex vivo porcine model of acute knee injury [127]. Translating to an in vivo model, Wu et al. performed a murine study analyzing PTOA progression comparing treatment with systemic bisphosphonate injection versus mechanical compression of the joint [145]. The mechanical loading was administered in a squat-like position with knee flexion at 90 degrees and the force pressing down on the superior aspect of the thigh. The mechanical loading cycle consisted of a force of 1.8 N for 5 min at a frequency of 4 Hz with treatment occurring 5 days per week for 3 weeks. They found that at the early timepoint (4 weeks), the compression group, bisphosphonate group, and combination treatment group (bisphosphonate + compression) all had less cartilage destruction, fewer osteoclasts, and higher BV/TV and bone mineral density. Finally, at 8 weeks, it was found that the combination treatment had a synergistic effect on protecting hyaline cartilage and proteoglycans, solidifying that mechanical compression can provide further therapeutic value than bisphosphonates alone.

A similar study analyzed joint healing and adipose-derived stem cell (ASCs) migration after treatment with ASC injection and mechanical compression (1.0 N for 6 min at 5 Hz for 2 weeks) [146]. They found that joint loading provided increased ASC migration at the joint and accelerated repair of the OA-damaged sites. Similar mechanical loading models have also found relief of abnormal remodeling of subchondral bone and osteoclast inhibition [147,148]. Other pre-clinical models have used an aerobic rehabilitation strategy for testing mechanical load to treat OA. Hao et al. used a mouse OA model to study the impact of aerobic treadmill walking (30 min per day 5 days a week for 4 weeks) on disease progression [126]. They found that both BW walking and supported BW walking (60% of BW) alleviated the OA-induced changes; however, the supported BW walking better maintained cartilage integrity and attenuated subchondral bone loss.

The therapeutic effects of mechanical loading found in previous studies can be used to define the Goldilocks zone and have potential to be translated to clinical treatment in a similar way. The cyclical duration of the mechanical compression would stay the same, but the force would have to be adjusted according to the patient’s body weight. For example, a 70 kg patient may need a force of around 100–200% BW (approximately 700–1400 N) to provide adequate stimulus. Based on our biomechanical analysis of rehabilitation exercises, this external force would best be applied at a knee flexion of 80–90 degrees. This angle would best translate to tibiofemoral compression and minimize shearing force. The ankle joint could be slightly dorsiflexed to prevent anterior sliding during compression. Additionally, following traumatic injuries, patients may benefit from treadmill walking with a reduced body weight. This can be performed in a low gravity G-trainer treadmill that provides lower body positive pressure and reduces effective weight. Takacs et al. studied the difference between normal treadmill walking and supported treadmill walking (87.6% BW) using a G-trainer [149]. They found that patients using the G-trainer had less pain, but there was no significant change in knee osteoarthritis outcome scores. However, this study only performed two exercise sessions per patient and more sessions are necessary to see OA regression. Further clinical studies will need to be conducted to identify the optimal parameters for mechanical compression, including compressive force, frequency, length of treatment, and time to begin treatment.

## 4. Biomechanical Strategies for Prevention and Treatment of KOA and PTOA

### 4.1. Preventing PTOA Development by Monitoring Joint Loading

As described before, joint loading surpassing a critical threshold (i.e., 18 MPa) may cause articular cartilage injury. Recent advances in technologies of wearable sensors and artificial intelligence may enable us to real-time monitor joint forces in sport and daily activities [150,151], which can allow us to evaluate the risk of join injury and prevent PTOA development by early effective interventions, such as therapeutic joint rehabilitation. Additionally, by using the technologies of real-time monitoring of joint forces, future studies can elucidate the loading conditions for articular cartilage injuries that lead to PTOA.

### 4.2. Mitigating OA by Reducing Abnormal Joint Biomechanics

Venturing into the sphere of OA management, our focus now shifts to the exploration of current investigational therapies leveraging mechanical loading principles, a topic integral to our prior discussions. With aging, many people develop weak quadriceps and subsequent knee adduction moment (KAM) while walking. The KAM has been shown to increase tibiofemoral compression force by up to 50% [111]. After years of improper biomechanics, many patients suffer from medial compartment OA due to this increasing varus malalignment [129]. There are currently few treatment modalities used to provide symptomatic relief and offload the compression from the medial compartment.

Wedge-shaped orthotics that inhibit knee adduction are often used to control pain in medial compartment OA. While it is proven that these insoles can alter the knee adduction moment by 13–15%, the clinical magnitude of this intervention’s effect is unknown and there is insufficient evidence to suggest it prevents disease progression [101,102]. Additionally, it is unknown if these orthotics have any negative effects on ankle or hip alignment, leading to further problems.

Other options include gait modification. Gait walking studies can determine the level of knee adduction and displacement. Through patient education and physical therapy, these abnormalities can be corrected, resulting in proper alignment and mechanical compression at the joint. Three modifiers have been studied. The ipsilateral trunk lean has been the most effective, reducing the KAM by up to 65% [129]. This method involves leaning toward the affected side and shifting the center of mass to the lateral compartment. The medial-thrust gait pattern has been effective as well, reducing the KAM by up to 50–55% [104]. This technique intentionally places the step more lateral, requiring the knee to thrust medially to maintain balance. Finally, fanning the toes outward has been proven to reduce KAM by 7–11% [104].

Knee bracing is another popular option for medial compartment OA. A valgus knee brace can be used to prevent varus movement by 25–34% [102]. This has been shown to reduce pain and increase functionality. One study looked at a valgus and external rotation brace with and without wedge foot orthoses. They concluded that the valgus and external rotation brace was biomechanically most effective, and the addition of foot orthoses provided unnecessary discomfort with minimal benefit [103]. Despite the biomechanical realignment, these braces are expensive and require strict compliance to prevent disease progression.

### 4.3. Treating KOA by Application of Controlled Joint Loading Defined in the Goldilocks Zone

The therapeutic effects of mechanical loading on chondrocytes were observed for specific loading conditions (i.e., specific combinations of loading magnitude, frequency, and duration) [141,142,143,144] as shown in the Goldilocks zone. Robotic rehabilitation systems can be used to provide precise and customized therapy tailored to the individual needs of patients [127,152]. Robotic technology allows for accurate control of joint loading and movement defined in the Goldilocks zone. These systems offer consistent and repetitive rehabilitation sessions, ensuring that patients receive a standardized level of care.

Since OA reduces the mechanical properties of AC, the magnitudes of therapeutic loading in the Goldilocks zone may need to be reduced. In research conducted by He et al., it was shown that lower levels of mechanical loading hindered the progression of cartilage destruction, subchondral bone alterations, and inflammation within OA joints [51]. There has been increasing interest in the use of low-resistance/gravity and water-based activities to optimize the loading stress on the joint. While previous studies also emphasize the importance of muscle strength training with KOA, their findings surprisingly find no advantage in utilizing low-resistance gravity training in the management of KOA in an elderly population [105]. In water training, however, significant results have emerged demonstrating the decreased mechanical load incurred during water training contributes to improved functional and pain scores in patients with KOA [106,108]. Although limited studies have explored low-resistance training in a preventative capacity for KOA, the existing evidence and mechanical underpinnings of KOA suggest a potential for utilizing it in this manner.

### 4.4. Future Directions and Conclusion

In contemplating future directions for research, it is evident that while there are ongoing and promising studies focused on harnessing the principles of mechanical loading, the potential for advancements is vast. It is widely acknowledged that the pathogenesis of OA is intricately linked to inflammatory pathways, often initiated by aberrant mechanotransduction pathways. Therefore, there is a compelling need to amplify our efforts in this realm, as it holds the potential to mitigate or prevent OA entirely, negating the necessity for interventional therapeutics that may carry unwanted side effects.

Moreover, there is a call to delve deeper into isolated, single-variable analysis to delineate a precise mechanical load range capable of inducing chondrocyte damage or injury. Our discussions within this review have underscored the multifactorial nature of knee joint injury and the progression of KOA, which significantly increases the difficulty in doing such univariate analysis. To make significant strides in this field of research, however, it is imperative to first pinpoint a specific mechanical load that triggers damage. Once identified, this established range can then be applied, adapted, and individualized to accommodate various factors and describe changes induced under diverse conditions. Such a nuanced approach holds the potential to catalyze a boom in research within the field, driving our understanding of knee joint health to new heights.

## Figures and Tables

**Figure 1 bioengineering-11-00110-f001:**
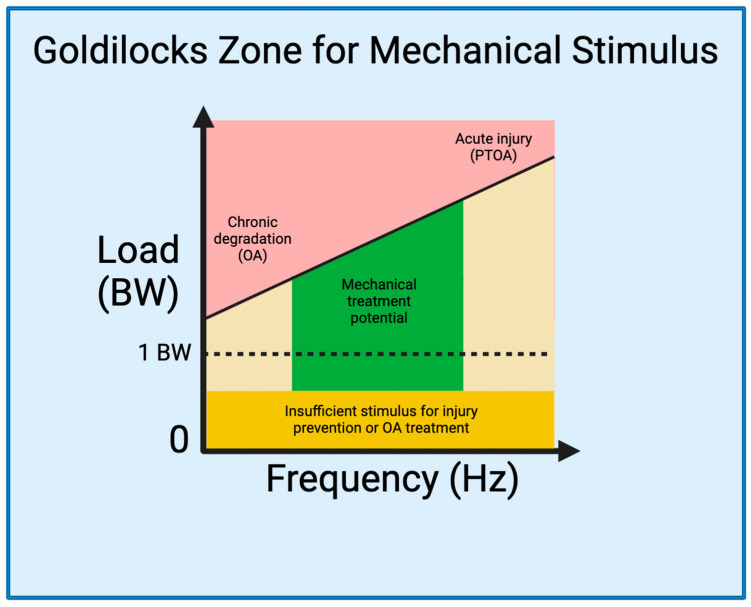
The Goldilocks zone for mechanical stimulation.

**Figure 2 bioengineering-11-00110-f002:**
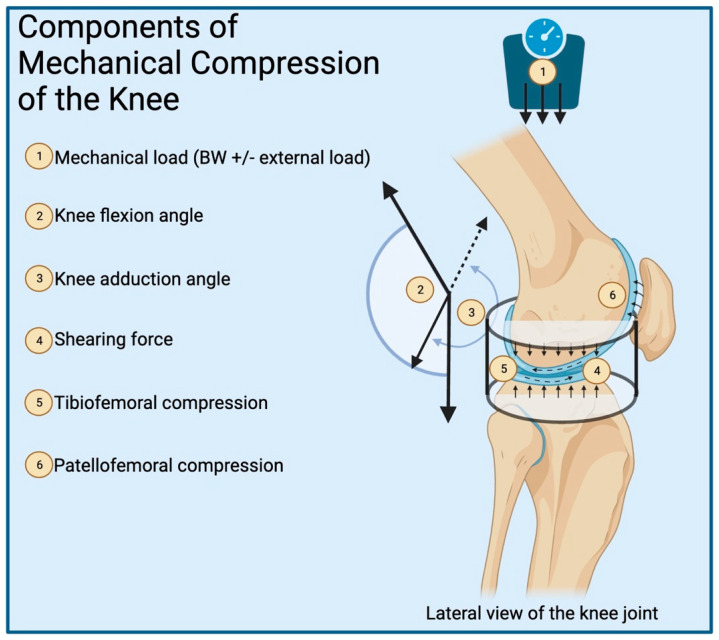
Mechanical compression of the knee.

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
