# Peer review of "Finding the Goldilocks Zone of Mechanical Loading: A Comprehensive Review of Mechanical Loading in the Prevention and Treatment of Knee Osteoarthritis"

_bioengineering, 2024, doi:10.3390/bioengineering11020110_

Round 1

Reviewer 1 Report

Comments and Suggestions for Authors

General Comments:

You review is well written and does a great job of covering the knee and the problems that affect it. I only have a few technical issues to bring to your attention.

Specific Recommendations:

Page 1, line 10 of the Introduction – You defined OA in your Abstract but not hear. However, you did define KOA here and in the Abstract. To be consistent you need to define both or used the definition from the Abstract for both.

“…greatly contributes to the urgency of addressing OA as a public health concern that de-“

Page 5

Line 9 (first full paragraph – Some of your readers will not know that MPa stands for “Microscopic Polyangiitis”. It should be defined.

Line 20 (Table 1) – I question the need for Table 1. Your references cover this.

Page 9, line 16 – You give the figure number at the top of this page so there is no need to repeat it here. “Goldilocks zone could prevent joint injury and damage, whereas exhibiting opposite effects outside the zone (Figure 1).”

Page 10 - The usage of “Introduction” for the third section (3. Biomechanics of Rehabilitation Exercises: 3.1. Introduction) is inappropriate. It does not match the other sections. I would suggest that you change it to “Rehabilitation Technics.”    

Author Response

  1. Page 1, line 10 of the Introduction – You defined OA in your Abstract but not hear. However, you did define KOA here and in the Abstract. To be consistent you need to define both or used the definition from the Abstract for both. “…greatly contributes to the urgency of addressing OA as a public health concern that de-“

Response: It has been corrected as suggested.

  1. Page 5, Line 9 (first full paragraph – Some of your readers will not know that MPa stands for “Microscopic Polyangiitis”. It should be defined.

Response: MPa stands for the pressure unit of megapascals. It has been defined and added in the manuscript and the abbreviations table.

  1. Page 9, line 16 – You give the figure number at the top of this page so there is no need to repeat it here. “Goldilocks zone could prevent joint injury and damage, whereas exhibiting opposite effects outside the zone (Figure 1).”

Response: It has been corrected.

 Page 10 - The usage of “Introduction” for the third section (3. Biomechanics of Rehabilitation Exercises: 3.1. Introduction) is inappropriate. It does not match the other sections. I would suggest that you change it to “Rehabilitation Technics.”    

Response:  Thank you for the suggestion. It has been revised as suggested.

Reviewer 2 Report

Comments and Suggestions for Authors

a very well done review of an intriguing issue

the language is correct and the references are complete

the paper puts a light on physiopathology of OA e PTOA in an excellent manner 

Author Response

Thank you for the constructive comments.

Reviewer 3 Report

Comments and Suggestions for Authors

This is a very nicely written and organized literature review related to this topic area and I found it very interesting overall, however I do not find that it meets the criteria to be characterized as a "systematic review" as there was no indication of a systematic approach to selecting and analyzing or assessing the cited literature.  I think if there is a category for a general literature review, that would be fine, but I don't think it really should be called a systematic review. 

Author Response

Agree. The manuscript has been categorized as a general literature review in the system.

Reviewer 4 Report

Comments and Suggestions for Authors

General information about the article:

The paper Finding the Goldilocks Zone of Mechanical Loading: A Comprehensive Review of Mechanical Loading in the Prevention and Treatment of Knee Osteoarthritis provides a thorough examination of the intricate relationship between mechanical factors and the pathogenesis of knee osteoarthritis (KOA) and post-traumatic osteoarthritis (PTOA). The authors skillfully delve into the crucial role of mechanotransduction in inflammatory responses, highlighting its potential for application in the prevention and rehabilitation of osteoarthritis (OA).

One of the key contributions of this review is the introduction of the concept of the "Goldilocks zone." This term encapsulates the idea that maintaining a delicate balance of mechanical loading is essential for preserving optimal knee joint health. The paper effectively argues for the importance of avoiding both inadequate and excessive mechanical loading, emphasizing the need for precision in therapeutic approaches.

The discussion extends beyond theoretical considerations, incorporating practical insights by exploring the biomechanical loading associated with various rehabilitation exercises. This pragmatic approach holds promise for physicians managing KOA and PTOA, as well as athletic training staff seeking to strategically plan athlete loads to minimize the risk of joint injury.

The synthesis of current literature findings adds depth to the paper, offering a multifaceted analysis of the contributing factors to KOA and PTOA. By emphasizing the integration of these concepts, the authors encourage a holistic understanding of the conditions, fostering a foundation for future research endeavors.

The paper concludes by advocating for further investigation into the body's physiological responses to mechanical stimuli in OA management. This call to action underscores the potential of harnessing the body's own mechanisms to optimize therapeutic interventions. Overall, the paper not only consolidates existing knowledge but also propels the field forward by illuminating avenues for future exploration and application.

The paper is generally well-written but will require minor revisions before proceeding further. I have included detailed comments below.

Minor comments:

The incidence of osteoarthritis is influenced by many factors, such as work, sports participation, musculoskeletal injuries, obesity and gender. Information about this, along with the necessary literature, should be added in the introduction preferably in the 3rd paragraph. Authors may find some useful information in the works: DOI 10.3390/app11041552; DOI 10.3390/app10238312; doi:10.35784/acs-2022-14;

Is marking in yellow on page 3 necessary ? Please correct the formatting of the text.

The purpose of the work should be more clearly outlined. Please rewrite the last paragraph of the introduction and increase the emphasis on the purpose of this review. 

I think in Table 1 the source should be listed right after the author's name. Please merge these columns and rebuild Table 1. This will allow it to be smaller.

The title of Figure 1 is far too long. Please move part of the title to the description in the text directly below the figure. The same goes for Figure 2. Please rebuild the text.

In the third paragraph of Section 3.2, please define the assumptions of each kinetic chain in more detail, along with the necessary literature. Authors can find useful information in the following papers: https://doi.org/10.1016/j.pmrj.2011.02.021     DOI 10.3390/s22103765   

The text also needs to be checked for formatting, especially in terms of citing tables and figures. Please adapt your work to the requirements of the journal.

After making appropriate corrections and completing the literature, the work can be accepted for publication.

Author Response

  1. The incidence of osteoarthritis is influenced by many factors, such as work, sports participation, musculoskeletal injuries, obesity and gender. Information about this, along with the necessary literature, should be added in the introduction preferably in the 3rd paragraph. Authors may find some useful information in the works: DOI 10.3390/app11041552 (Krakowski); DOI 10.3390/app10238312 (Krakowski); doi:10.35784/acs-2022-14 (Karpinski);

Response: Thank you so much for bringing these sources to our attention. They have been added in support of the multifaceted nature of OA progression within our introduction.

  1. Is marking in yellow on page 3 necessary ? Please correct the formatting of the text.

Response: It has been corrected.

  1. The purpose of the work should be more clearly outlined. Please rewrite the last paragraph of the introduction and increase the emphasis on the purpose of this review. 

Response: Thank you for this suggestion. We have re-worded the conclusion of our introduction to be clearer about the purpose of our review.

  1. I think in Table 1 the source should be listed right after the author's name. Please merge these columns and rebuild Table 1. This will allow it to be smaller.

Response: Table 1 has been corrected and re-formatted as suggested.

  1. The title of Figure 1 is far too long. Please move part of the title to the description in the text directly below the figure. The same goes for Figure 2. Please rebuild the text.

Response: Thank you for this suggestion. We separated and reformatted the figure title from the figure legend, in order to make it clearer for readers.

  1. In the third paragraph of Section 3.2, please define the assumptions of each kinetic chain in more detail, along with the necessary literature. Authors can find useful information in the following papers: https://doi.org/10.1016/j.pmrj.2011.02.021     DOI 10.3390/s22103765   

Response: We thank the reviewer for the recommendation on additional sources and agree that they contribute to our work. The third paragraph of section 3.2 has been revised to cover kinetic chain in additional detail.

  1. The text also needs to be checked for formatting, especially in terms of citing tables and figures. Please adapt your work to the requirements of the journal.

Response: Thank you for your feedback and for your suggestions. We reviewed our formatting and made corrections to fit the journal’s requirements.

 After making appropriate corrections and completing the literature, the work can be accepted for publication.

Response: We thank the reviewer for his detailed feedback.

Reviewer 5 Report

Comments and Suggestions for Authors

It is an interesting review, and the topic knee osteoarthritis is very important nowadays. The paper is well structurized, clearly written. Abstract is comprehensive, and encourages to read the whole text. 

I have only few notes:

table 12 - Schroder et al (47) - there is lack of study description

part 3.3  (squat)- I would like to read a little bit about patellofemoral forces also. 

Author Response

  1. table 12 - Schroder et al (47) - there is lack of study description

Response: Thank you so much for bringing this to our attention. It has been corrected.

  1. part 3.3  (squat)- I would like to read a little bit about patellofemoral forces also. 

Response: Thank you for the suggestion, we added additional information about patellofemoral forces within part 3.3.

Round 2

Reviewer 3 Report

Comments and Suggestions for Authors

thank you for changing the general description, this manuscript is fine as a general review article.